# Increased Cuticle Waxes by Overexpression of WSD1 Improves Osmotic Stress Tolerance in *Arabidopsis thaliana* and *Camelina sativa*

**DOI:** 10.3390/ijms22105173

**Published:** 2021-05-13

**Authors:** Hesham M. Abdullah, Jessica Rodriguez, Jeffrey M. Salacup, Isla S. Castañeda, Danny J. Schnell, Ashwani Pareek, Om Parkash Dhankher

**Affiliations:** 1Stockbridge School of Agriculture, University of Massachusetts Amherst, Amherst, MA 01003, USA; abdull76@msu.edu (H.M.A.); jessicarodri@umass.edu (J.R.); 2Biotechnology Department, Faculty of Agriculture, Al-Azhar University, Cairo 11651, Egypt; 3Department of Plant Biology, Michigan State University, East Lansing, MI 48824, USA; schnelld@msu.edu; 4Department of Geosciences, University of Massachusetts Amherst, Amherst, MA 01003, USA; jsalacup@umass.edu (J.M.S.); isla@cns.umass.edu (I.S.C.); 5Stress Physiology and Molecular Biology Laboratory, School of Life Sciences, Jawaharlal Nehru University, New Delhi 100067, India; ashwanip@mail.jnu.ac.in

**Keywords:** cuticular waxes, drought, salinity, stress tolerance/adaptation, gene expression, wax loading

## Abstract

To ensure global food security under the changing climate, there is a strong need for developing ‘climate resilient crops’ that can thrive and produce better yields under extreme environmental conditions such as drought, salinity, and high temperature. To enhance plant productivity under the adverse conditions, we constitutively overexpressed a bifunctional *wax synthase/acyl-CoA:diacylglycerol acyltransferase* (*WSD1*) gene, which plays a critical role in wax ester synthesis in Arabidopsis stem and leaf tissues. The qRT-PCR analysis showed a strong upregulation of *WSD1* transcripts by mannitol, NaCl, and abscisic acid (ABA) treatments, particularly in *Arabidopsis thaliana* shoots. Gas chromatography and electron microscopy analyses of Arabidopsis seedlings overexpressing WSD1 showed higher deposition of epicuticular wax crystals and increased leaf and stem wax loading in WSD1 transgenics compared to wildtype (WT) plants. WSD1 transgenics exhibited enhanced tolerance to ABA, mannitol, drought and salinity, which suggested new physiological roles for WSD1 in stress response aside from its wax synthase activity. Transgenic plants were able to recover from drought and salinity better than the WT plants. Furthermore, transgenics showed reduced cuticular transpirational rates and cuticle permeability, as well as less chlorophyll leaching than the WT. The knowledge from Arabidopsis was translated to the oilseed crop *Camelina sativa* (L.) Crantz. Similar to Arabidopsis, transgenic Camelina lines overexpressing WSD1 also showed enhanced tolerance to drought stress. Our results clearly show that the manipulation of cuticular waxes will be advantageous for enhancing plant productivity under a changing climate.

## 1. Introduction

Climate change is having a substantial impact on agricultural production worldwide and will have serious consequences for global food security [1]. As a result of the changing climate, elevated temperatures, drought, salinity, heavy metals contamination, and other types of damaging weather are all expected to take their toll on crop yield and quality [2]. Today’s agriculture is facing a major crisis in order to produce enough food for the growing world population. By 2050, farmers are expected to increase the global food production by 70–110% to feed the population of more than 9 billion (Food and Agriculture Organization, FAO [3,4]), compounded with the dramatic loss and degradation of arable land due to increasing salinity and drought.

The growing food demand and the threat of heavy crop losses due to climate change will make feeding the world’s population in the future exceedingly challenging. Under the current climate change scenario, a major constraint for crop production is the availability of irrigation water. Drought stress is expected to limit the productivity of more than half of the earth’s arable land in the next 50 years, and competition for water between urban and agricultural areas will compound the problem [5,6]. Therefore, to ensure global food security, there is an urgent need to develop ‘climate-resilient crops’ that can thrive and produce better yields under extreme environmental conditions, such as drought, salinity, and high-temperature stress.

Cuticular waxes are hydrophobic layer coatings on the epidermis of the aerial plant bodies that act as the first barrier between the plants and the surrounding environments. These waxes are varied in structures and range from cutin polyester to intra and epicuticular waxes [7]. The chemical composition of cuticular waxes is also varied; they consist of solvent-soluble lipids composed of very-long-chain fatty acids (VLCFAs) and their derivatives (including aldehydes, alkanes, fatty alcohols, ketones, and wax esters), with a small portion of phenylpropanoids and triterpenoids [8]. The chemical and physical properties of cuticular waxes render them useful in differential physiological functions. On leaf surfaces, cuticular waxes can control non-stomatal water loss termed as residual transpiration and gas exchange [9,10,11], provide protection against UV radiation [7], prevent the attachment of pathogen spores and atmospheric pollutants, in addition to their roles in controlling plant interactions with insects, fungal, and bacterial pathogens [7]. The cuticular waxes help in maintaining equilibrium between the transpirational water loss and root water uptake by controlling the transpiration and uptake of polar solutes and regulating the exchange of gases and vapor [12].

Cuticular waxes are synthesized on the epidermal cells, and their synthesis is initiated by the elongation of C_16_-C_18_ fatty acids, which are produced in plastids to from VLCFAs (with carbon length up to >34) via the activity of fatty acid elongase (FAE) complex. FAE is comprised of four enzymes, β-ketoacyl-CoA synthase (KCS), β-ketoacyl-CoA reductase (KCR), β-hydroxyacyl-CoA dehydratase (HCD), and enoyl-CoA reductase (ECR), located in the endoplasmic reticulum (ER, [13]). The elongated VLCFAs are subsequently activated into VLCFA-CoAs by the activity of a long chain acyl-CoA synthetase (LACS), and then biochemically modified via the alkane or primary alcohol-forming pathways [14]. In the alkane-forming pathway, active VLCFAs are converted into alkanes via the activity of aldehyde decarbonylase (protein ECERIFERUM, CER) and cytochrome b5 complexes [15]. In the alcohol-forming pathway, the VLCFAs are reduced to primary alcohols via the activity of fatty acyl-CoA reductases, and the formed primary alcohols, along with acyl-CoA pools with 16 carbon atoms, are subsequently esterified via the activity of bifunctional WSD1 [13,16]. Like in Arabidopsis, several genes associated with cuticular wax biosynthesis have been identified and characterized in other crops, including maize, rice, and tomato [13,17,18,19,20], reflecting the need for cuticular wax biosynthesis as a critical biological process in all plant species.

Plants usually exhibit various physiological and biochemical changes in response to abiotic stresses. These changes include stomata closure, reduced photosynthesis efficiency and transpiration, and the synthesis of stress-related secondary metabolites [21]. Under drought stress conditions, plants have developed a strategy by which they can lessen the impact of drought stress by increasing the amounts of cuticular waxes. It was reported that Arabidopsis, tree tobacco (*Nicotiana glauca*), cotton (*Gossypium hirsutum*), and sesame (*Sesamum indicum*) plants produce approximately 100%, 100%, 70%, and 30% greater amounts of cuticular wax under drought stress conditions, respectively, compared to the plants grown under normal conditions [22,23,24,25]. More recently, it has been cited that the overexpression of several genes encoding transcription factors (TFs), which regulate the expression of cuticular wax biosynthetic genes, enhance drought tolerance in transgenic plants. For example, the amount of cuticular waxes has increased substantially, and drought resistance was observed in transgenics overexpressing (i) *WIN1*/*SHN1* gene encoding an AP2/EREBP family transcription factor in Arabidopsis [26], (ii) *Medicago truncatula WXP1* and *WXP2* genes in alfalfa (*Medicago sativa*) and Arabidopsis [27,28], (iii) *Brassica napus* (rapeseed) *BnLAS* gene encoding a GRAS transcription factor in Arabidopsis [29], (iv) MYB96 transcription factor in Arabidopsis [30], (v) *OsWR1* gene in rice (*Oryza sativa*, [31], and (vi) Arabidopsis *MYB96* gene in Camelina [13].

Furthermore, it was reported that, under drought conditions, the overexpression of Arabidopsis MYB96 activates the expression of wax biosynthetic genes and increases the accumulation of cuticular waxes [30]. Among the genes that were shown to be up-regulated via the activity of MYB96 was *WSD1*, which is evidenced to play a critical role in wax ester synthesis in Arabidopsis stem tissues [30,32]. Whether increasing the transcriptional activity of *WSD1* would affect the amount of cuticular waxes, which can contribute to providing tolerance to drought stress, is still less clear. In this study, the Arabidopsis wax synthase, *WSD1* (AT5G37300) gene, encoding a bifunctional WSD1, was constitutively overexpressed in Arabidopsis and Camelina under the control of the *CaMV 35S* promoter. Transgenics were generated, tested for increased deposition of epicuticular waxes in leaf and stem surfaces using gas chromatographic-mass spectrometry (GC-MS) techniques and scanning electron microscopy (SEM), and finally were evaluated for their response to drought and osmotic stresses.

## 2. Results

### 2.1. Differential Regulation of WSD1 Transcripts under Abiotic Stress Conditions

To determine whether the expression of *WSD1* gene is induced/regulated by abiotic stresses including mannitol, salt, and the stress hormone ABA, qRT-PCR was performed to measure it’s relative expression in Arabidopsis shoot and root tissues. Our results indicated that the expression of *WSD1* was induced rapidly upon exposure to mannitol, NaCl, and ABA treatments, particularly in shoots (Figure 1B,D,F), and no induction was detected in roots (Figure 1A,C,E). When treated with 100 mM mannitol, the *WSD1* transcript level was elevated by more than 2.5 folds. It peaked within 6 h and then gradually declined (Figure 1B). Furthermore, the expression of *WSD1* was also strongly up-regulated up to 10-folds by NaCl treatment (150 mM) within 6 h and was continuously elevated for up to 24 h (Figure 1D). In addition, the expression of *WSD1* was elevated up to 8-folds in response to ABA (1 μM), with its expression peaking within 12 h and then declining almost to normal levels after 24 h (Figure 1F). These findings suggested that the strong induction of the Arabidopsis *WSD1* gene with exposure to abiotic stresses probably contributes to regulating physiological roles in stress response, aside from its wax synthase activity.

### 2.2. Generation of Arabidopsis Transgenic Plants Overexpressing CaMV35S::WSD1 Construct

In order to study the physiological roles of the cuticular wax biosynthetic genes under abiotic stresses, transgenic Arabidopsis plants overexpressing the *WSD1* gene under the control of the constitutive *CaMV 35S* promoter were generated (Appendix A). T1 seeds were screened using red fluorescence of the vector *pCAMBIA-DsRed* and further confirmed by selecting the transformants on ½ MS medium containing 20 mg L^−1^ hygromycin-B antibiotic. The transgenic seedlings were grown to raise further generations. Constitutive overexpression of WSD1 in the shoots and roots of Arabidopsis plants grown on MS medium was confirmed using qRT-PCR (Appendix A). We find that the relative expression of *WSD1* transcripts was 5–40 fold and 10–160 fold higher in shoots and roots, respectively, of the transgenic plants compared to non-transgenic WT plants. The qRT-PCR results confirmed that the *WSD1* gene was constitutively overexpressed at higher levels in transgenics, and T3 homozygous transgenic plants were further used in the subsequent analyses.

### 2.3. Epicuticular Wax Quantity and Composition Is Altered in WSD1 Overexpressing Lines

To elucidate whether the constitutive overexpression of WSD1 expression elevated the amount of waxes and/or altered their composition, GS-MS coupled with GC-FID and scanning electron microscopy (SEM) were used to profile the cuticular waxes deposited in Arabidopsis leaf and stem tissues. Total wax content and wax composition were analyzed. SEM imaging of Arabidopsis leaf and stem tissues clearly indicated that the deposition of epicuticular wax crystals was increased on the transgenic leaf and stem surfaces relative to their amount in WT leaf and stem (Figure 2).

Furthermore, the measurement of cuticular wax content and composition in leaf and stem tissues of Arabidopsis WT and WSD1 transgenics by GC-FID and GC-MS, respectively, also supported the SEM imaging data that the total wax load was increased by 40–70% in leaf (Figure 3A) and by 15–30% in stem (Figure 4A) tissues in WSD1 transgenics as compared to WT plants. Additionally, the impact of the constitutive overexpression of AtWASD1 on altering the cuticular wax composition in Arabidopsis leaf tissues was significant (Figure 3A). Leaf alkanes, fatty alcohols, sterols, aldehydes, ketones, and fatty acids were increased dramatically in WSD1 overexpressing lines compared to WT (Figure 4A). Alterations in the contents of alkanes and fatty alcohols were the most prominent changes. The levels of alkanes were elevated in WSD1 leaf (up to 77–130%) compared to their levels in WT, while the fatty alcohols content increased by 72–141% more in WSD1 leaf than in WT leaf. Notably, most of the changes in the quantity of leaf wax in WSD1 lines could be attributed mostly to the 74–136% increases of the C_29_ and C_31_ *n*-alkanes, and 42–150% increases in C_28_ and C_32_ fatty alcohols (Figure 3B). Unlike in Arabidopsis leaf, the impact of AtWSD1 overexpression on the cuticular wax amount and composition in stem tissues was less obvious. The total wax loading did not change in the stem of transgenic plants, and the levels of alkanes, fatty alcohols, ketones, and fatty acids remained largely unaffected (Figure 4A,B).

### 2.4. WSD1 Overexpressing Lines Exhibited Increased Abiotic Stress Tolerance in MS Media

To analyze whether the increased cuticular wax contents and altered composition in WSD1 overexpressing lines conferring tolerance to various osmotic stresses, WSD1 transgenics and WT seeds were germinated in ½ MS medium supplemented with ABA (1 μM), mannitol (100 mM), and NaCl (75 mM, Figure 5). Seedlings phenotypes were observed (Figure 5A) and total plant biomass and root length data were recorded from 21-day-old seedlings (Figure 5B–I). In the absence of stress agents, transgenic plants growth was similar to that in WT plants with no difference in root length and total plant biomass (Figure 5B,C). When grown on ½ strength MS medium containing ABA, WSD1 transgenic lines grew much better than the WT plants and exhibited ~1.5-fold longer roots and attained ~2.5-fold higher plant biomass compared to WT plants (Figure 5D,E). Furthermore, WSD1 transgenic plants treated with mannitol attained slightly longer roots, whereas the plant biomass significantly increased (~1.5-fold) in WSD1 transgenics compared to WT plants (Figure 5F,G). WSD1 transgenic plants, when treated with salt (75 mM NaCl), the root length and shoot biomass of only line #8 were significantly increased as compared to that in WT plants (Figure 5H,I), however, other lines also had slightly, but not significantly, longer roots under the assay conditions. Collectively, the transcriptional induction of WSD1 under ABA, Mannitol and salt treatments, and the resistant phenotype observed in WSD1 overexpressing lines under these treatments indicated the critical physiological roles that WSD1 may play in drought, salt and ABA-mediated stress responses.

### 2.5. WSD1 Overexpressing Lines Exhibited Increased Drought and Salinity Stress Tolerance in Greenhouse Conditions

Since WSD1 is involved in cuticular wax biosynthesis and accumulation [32], we examined more directly whether increasing wax deposition by constitutively overexpressing WSD1 in Arabidopsis is linked to drought response in soil under greenhouse conditions. WT and WSD1 transgenic plants were subjected to drought stress by withholding water for 12 days. The WSD1 overexpressing lines and WT plants were grown under either normal or drought conditions, and the leaves of 4-week-old plants were subjected to water loss and chlorophyll leaching assays as indicators for changes in epicuticular transpiration and permeability, respectively. The transgenic WSD1 plants exhibited a strong drought-tolerant phenotype after water was withheld for 12 days, and the transgenic plants were able to recover from drought stress better than the WT plants (Figure 6A). The delayed wilting and faster recovery phenotypes of WSD1 transgenic plants were correlated with a reduction in water loss from the leaf tissue, indicating a reduction in cuticular transpirational rates (Figure 6B). The leaf water loss assay showed that the cuticular transpiration occurred more slowly in the WSD1 transgenic leaf compared to that in the WT leaf. Within one hour of leaf drying, the transgenic leaf was able to maintain ~60% of its water content, while the WT leaf lost ~90% of its water content (Figure 6B).

It has been shown that changes in cuticular wax quantity and composition can affect epidermal properties, i.e., epidermal permeability [11,30,33]. To support our findings that these changes, along with the alteration in epidermal properties, have caused the drought phenotype, we performed a chlorophyll leaching assay in Arabidopsis plants. WSD1 overexpressing plants showed reduced chlorophyll leaching compared to WT leaves, possibly due to lower cuticle permeability (Figure 6C).

Additionally, we examined whether the increased accumulation of cuticular waxes via the overexpression of WSD1 could also confer resistance to salinity stress. WT and transgenic plants were grown in soil treated with salt (100 mM NaCl) and the plants’ response to salinity was recorded (Figure 7). The plants were grown in the greenhouse under normal watering conditions for four weeks and then divided into two groups. One group of plants was continuously watered normally, and the other group of plants was subsequently watered with 100 mM NaCl-containing water for 6 days and then allowed to recover for another 6 days (Figure 7A). The leaves of WT exhibited severe bleaching and withering, whereas only a few leaves on transgenic plants exhibited slight wilting. After recovery, these WSD1 transgenics and WT were grown for another two weeks. Transgenic lines recovered well and showed better growth, produced flowers and siliques, whereas WT plants showed severe damage and recovered more slowly and were stunted with no or fewer flowers and siliques (Figure 7B). Since the link between the physiological roles of wax biosynthesis and salinity stress response, adaptation, and tolerance have not been investigated, we suggest further in-depth research utilizing WSD1 overexpressing lines to understand the interaction between cuticular waxes and salinity stress response.

### 2.6. Translating the Knowledge from Arabidopsis to the Oilseed Crop Camelina

Both Arabidopsis and Camelina are relatives belonging to the Brassicaceae family. Given the notion that many aspects of plant development are controlled in conservative modes among different plant species, we herein attempted to transfer the knowledge from Arabidopsis to the oilseed crop Camelina, through testing whether expressing WSD1 would have a similar impact on the physiology of Camelina plants under stress conditions, i.e., drought. The *CaMV35S::WSD1* construct was introduced into the Camelina genome, and transgenic T_1_ seeds showing DsRed fluorescence were selected and the resulting T_1_ plants were generated (see Materials and Methods section). The transgenic T1 plants were screened for the overexpression of the *WSD1* gene into Camelina using qRT-PCR analyses. The results indicated that WSD1 was successfully expressed in Camelina tissues and its overexpression was confirmed in transgenic Camelina seedlings (Appendix A). T3 homozygous transgenic plants were further used in the subsequent analyses.

As aforementioned, WSD1 expression has led to a drought tolerance in Arabidopsis plants grown in soils under drought stress. In a similar way, to assess the response of Camelina for plant growth and seed attributes under drought stress, both WT and WSD1 expressing plants (4 weeks old) were subjected to two drought stress levels, i.e., control (100% field capacity, FC) and water stress (50 and 75% FC). After 2 and 4 weeks of drought stress treatment, the plants were monitored for signs of wilting, and photographs were taken. As shown in Figure 7, planting with 50 and 75% reduction in soil water content for 2 or 4 weeks caused the WT plants to wilt faster, with considerably smaller and poorly developed leaves with yellowish appearance and less branching, whereas WSD1 plants exhibited a relatively delayed wilting, with well-developed leaves and more panicles, which indicate a strong drought-tolerant phenotype (Figure 8A). At these drought conditions, all WSD1 lines attained significant increases in biomass (Figure 8B) and panicles number per plant compared to their relative WT (Figure 8C).

Since drought conditions substantially limit seed production and quality, we examined whether expressing WSD1 in Camelina could have a positive impact on seed attributes under drought stress. The size and weight of WSD1 transgenic and WT seeds were measured. Under the well-watering condition, there are no changes observed in seed size or seed yield between WT and WSD1 transgenics. Nonetheless, under water stress in both 50% FC or 75% FC, all the plants, including WT, showed a reduction in seed yield; however, the WSD1 expressing Camelina lines showed significantly higher per plant seed yield compared to WT (Figure 8D). These increases in seed yields could be attributed to the increases in the number of panicles per plant observed in WSD1 lines under drought conditions, as shown in Figure 7C. Relatively, it was obvious that the poorly-developed panicles in response to drought have lowered the amount of seeds produced per plant and therefore allowed relatively bigger seeds to be formed more than that observed under the well-watering condition as a result of re-direction of resources allocation. However, the impact of drought stress was minimal on WSD1 expressing seeds, as these seeds exhibited significantly bigger sizes (indicated from the weight of 100 seeds) under drought conditions, relative to the that in WT seeds (Figure 8E). Under drought stress conditions, the WSD1 expressing also had a significantly higher number of panicles compared to WT plants (Figure 8C). Since the WSD1 encodes a bifunctional wax synthase/acyl-CoA:diacylglycerol acyltransferase, we were interested to assess whether WSD1 expression can also affect the total seed oil contents in Camelina. Transgenic Camelina lines showed slightly but significantly higher (5.0–8.5%) oil contents in seeds in comparison to WT seeds (Appendix A) under normal growth conditions. Since the purpose was to test whether the WSD1-associated phenotypes observed in Arabidopsis could be repeated in Camelina in a proof of concept perspective, the physiological, molecular, and biochemical aspects of WSD1 expression into Camelina will be the subject of further study and will be published elsewhere in the near future.

## 3. Discussion

Several studies have shown that the overexpression of TFs regulating the genes involved in the synthesis of cuticular waxes increased tolerance to drought stress [13,26,27,28,29,30,31]. The information regarding the WSD1 overexpression and its impact on drought and other stress tolerance in plants is limited. In the current study, we overexpressed the *WSD1* gene, under the control of *CaMV 35S* promoter in Arabidopsis and Camelina, which is evidenced to play a critical role in wax ester synthesis in Arabidopsis [32]. The resulting transgenic Arabidopsis lines demonstrated the significant higher deposition of epicuticular waxes and altered wax compositions. Furthermore, we examined more directly whether the increased wax deposition via increased WSD1 activity is linked to drought, salinity and ABA responses in both the MS media and soil under greenhouse conditions. WSD1 transgenic plants exposed to drought and salinity stresses in soil exhibited a strong tolerance phenotype, that the transgenic plants were able to recover from drought and salinity better than the WT plants. The delayed wilting, leaching, and drying in WSD1 transgenic plants were associated with (i) a reduction in cuticular transpirational rates, (ii) slower chlorophyll leaching which indicates lower cuticle permeability, (iii) increased deposition of epicuticular wax crystals, and (iv) increased leaf and stem wax loading, as compared to WT plants.

The qRT-PCR analysis of Arabidopsis plants indicated that *WSD1* expression is differentially regulated by mannitol, NaCl, and ABA treatments, particularly in Arabidopsis shoots (Figure 1). Upon exposure to these stresses, the transcript levels of *WSD1* increased, and the stress-dependent expression patterns of WSD1 indicated a possible correlation between the accumulation of wax esters, caused by WSD1 activity, and the associated abiotic stress response. In agreement with these findings, Seo et al. [30] and Patwari et al. [34] have also showed an up-regulation of *WSD1* transcripts in response to NaCl, cold, ABA, and drought treatments, suggesting potential roles for WSD1 under abiotic stress conditions.

Under normal environmental conditions, *WSD1* transcripts expression in Arabidopsis was detected primarily in flowers, in the top parts of the inflorescence stems, and in leaves, and its expression patterns and cellular localization are correlated with its suggested function as the major wax ester synthase of stems [24,32]. Moreover, WSD1 was up-regulated under several abiotic stresses, as shown in the current study, and as reported in [7,24,30], and its expression was suggested to be regulated by the activity of MYB96 [30] and MYB94 [7] TFs, which activate cuticular wax biosynthesis in response to environmental stresses including drought.

The up-regulation of *WSD1* transcripts observed under these stresses has encouraged us to study whether the Arabidopsis seedlings overexpressing WSD1 will produce stress-responsive phenotypes when grown on ½ MS media supplemented with stress agents. WSD1 transgenic lines showed strong resistance phenotypes in response to exogenous ABA, Mannitol, and NaCl compared to WT controls (Figure 5). Moreover, both Arabidopsis WT and WSD1 transgenic plants grown on soil were exposed to drought stress by withholding water, and the transgenics exhibited a stronger tolerance phenotype than that in WT plants (Figure 6). These results thus suggest new physiological roles for WSD1 in stress response aside from its wax synthase activity. Increased tolerance to salinity in the transgenic plants was also noticed, which not only has ionic stress caused due to excess Na+ ions, but an ‘in built’ component of osmotic stress. These results showed that a causal relationship between the amount of cuticle waxes and salinity tolerance is highly plausible. We speculate that higher levels of cuticular waxes reduce the extent of cuticular or residual transpiration and thus may improve water use efficiency (WUE) under hyperosmotic conditions. Our results are in corroboration with previous findings where similar relationship between the wax accumulation and residual transpiration has been reported in barley (*Hordeum vulgare* L.), leading to osmotic stress tolerance [11]. Furthermore, differential wax accumulation and increased stress tolerance has also been reported within the same plant when the young leaves and old leaves were analyzed. It would be interesting to study the role of *WSD1* gene in crop plants for increasing stress tolerance via restricting the residual transpiration using the tools of functional genomics.

To translate the results from Arabidopsis, we overexpressed the *WSD1* gene in the oilseed crop Camelina. Camelina has recently gained significant interest for developing as a dedicated biofuel crops [35,36] and increasing tolerance to abiotic stresses will enable it to grow on marginal lands and lands with undesirable characteristics while observing positive seed and oil attributes. Camelina transgenic plants showed drought tolerance potential, which could be associated with distinct physiological, biochemical, and molecular adaptations to drought condition. It looks promising that overexpression of WSD1 in Camelina plays critical roles in adapting the physiology of the plant to the changes in water stress levels in soils and allowing Camelina to produce better seed yields under such unfavorable conditions. However, further studies are needed in other crop species to investigate the roles that WSD1 plays under abiotic stress conditions.

Furthermore, the SEM, GC/MS and GC-FID analyses confirmed that expressing WSD1 in Arabidopsis has caused alteration to the cuticular wax amounts and composition. The SEM imaging (Figure 2) indicated the formation of additional wax structures on the surface of the adaxial side of the WSD1 leaves and in higher abundances in stems than that on the WT plants. The WT Arabidopsis stem surfaces are known to be covered by different wax structures, including rods, plates, crusts, filaments, and tubes, while Arabidopsis leaf surfaces usually produce a very few wax crystals, or none [37,38]. Moreover, the wax load and composition of Arabidopsis leaf and stem tissues of both WT and WSD1 transgenic lines indicated obvious changes in the amount and the composition of cuticular waxes, which could be attributed to the increased WSD1 activity. Overall, the wax profiles observed in the current study in Arabidopsis WT leaf and stem tissues were consistent with the previous reports [7,30,34,38], with *n*-alkanes and fatty alcohols predominating, and with relatively lower amounts of fatty acids, ketones, sterols, aldehydes, and triterpenes. Moreover, the total wax load in stems is significantly higher than that in leaves (~32 μg cm^−2^ in stem compared to ~1μg cm^−2^ in leaf). Most of the WSD1 transgenic plants subjected to the GC analysis observed increases in the total wax load, particularly in the transgenic leaf tissues, relative to WT leaves. This increase in total wax load was mostly obtained from the increase in the levels of C_29_ and C_31_ *n*-alkanes, C_28_-C_32_ fatty alcohols, and sterols. The total wax load was also increased, albeit not significantly, in the stem tissues of WSD1 transgenics, with some transgenic lines showing slight increases in the levels of C_29_ and C_32_ *n*-alkanes and C_26_ and C_28_ fatty alcohols (Figure 4).

It may also be noted that we were not able to quantify wax esters, the main products of WSD1 activity, due to some limitations in our GC analysis platform, probably because the wax esters accumulate in Arabidopsis leaf tissues in extremely low levels and were hard to be structurally determined. However, this limitation could be overcome, as Patwari and her co-authors in 2019 [34] were able to determine the wax ester content and composition in Arabidopsis leaf tissues using the thin layer chromatography (TLC) or solid-phase extraction on silica columns, coupled with the Q-TOF mass spectrometry, an approach which will be considered in our future research.

Furthermore, in the current study, the drought assay performed in both WT and WSD1 transgenic plants indicated that the stress-resistant phenotype observed in WSD1 transgenics could be directly caused by reduced cuticular transpirational rates and cuticle permeability. The water loss and chlorophyll leaching assays showed that the leaves of WSD1 transgenic lines could maintain their water content much longer and leaching of the chlorophyll much slower than their relative WT leaves. This reduction in water loss and chlorophyll leaching rates could be attributed to either lower permeability of the epidermis or decreased water loss through the stomata.

Interestingly, our finding of drought stress tolerance is in strong agreement with the opposite impact of the mutant alleles of WSD1 (designated *wsd1-1* and *wsd1-2*) on the leaf water content and leaf surface permeability reported in [34]. In this study, authors reported that *wsd1-1* and *wsd1-2* leaves lost their water content and leached chlorophyll faster than WT leaves, an impact that probably resulted from the increased number of stomata observed in the leaves of *wsd1-1* and *wsd1-2*. Relatively, under drought stress conditions, a leaf wilting and a severe reduction in relative water content (RWC) was observed in the *wsd1-1* and *wsd1-2,* indicating that WSD1 contributes to drought tolerance in Arabidopsis [34]. Our results further extend the role of WSD1 in providing strong tolerance to osmotic stresses caused by salinity and ABA exposure and our results in Camelina clearly indicate that the information from the model plant Arabidopsis can be successfully translated in crop plants.

There are many interpretations of the impact of WSD1 overexpression in conferring tolerance to drought stress seen in the current research. First, it could be the drought-response phenotype in WSD1 transgenics, which is caused by the direct impact of increased WSD1 activity on the accumulation of wax esters on stems and leaves. This increased wax deposition and composition could change the surface properties, probably through controlling the non-stomatal water loss by increasing the permeability of the cuticle as a consequence of more wax accumulation. The drought susceptibility phenotypes seen in *wsd1-1* and *wsd1-2* mutants [34] support this interpretation. Secondly, the drought response associated with increased cuticular waxes could contribute to the regulation of the *WSD1* gene and other wax biosynthetic genes via the activity of MYB94 and MYB96 TFs, under drought stress conditions [7,30]. Furthermore, our GC analysis results indicated that overexpressing WSD1 significantly increased the total wax load in Arabidopsis leaf, which mainly resulted from the increases in long chains *n*-alkanes (C_29_ and C_31_). Increasing the *n*-alkanes, and therefore the total wax load, under water deficit is a major mechanism by which many plants respond to drought conditions. For instance, it was reported that under water deficit, the Arabidopsis plants observed 93% increases in total wax load, and the major portion of these increases resulted from the accumulation of long chain *n*-alkanes (C_29_, C_31_, and C_33_ [23,39]). This induction of cuticle lipids was contributed to reduced cuticle permeability in Arabidopsis plants, an adaptation that is important for plant acclimation to subsequent water-limited conditions. In other studies, a decrease of the wax monomers, especially C_29_ *n*-alkanes, caused a decline in tolerance to drought stress in Arabidopsis [39,40].

Additionally, it was reported that under stress conditions, plants tend to reduce residual transpiration to improve their performance, and one way to do so is to increase the accumulation of cuticular waxes on the surface of the leaf [11]. According to Hasanuzzaman and his co-authors [11], four genotypes of barley contrasting in their tolerance to salinity were analyzed, and the results indicated a negative correlation between the residual transpiration and the osmolality, while a positive correlation between residual transpiration, the osmotic and leaf water potential, as well as the total cuticular wax content in leaf tissues, thus suggesting that residual transpiration could be an essential mechanism by which plants enhance WUE under stress conditions.

Additionally, the impact on *n*-alkanes, secondary alcohols, and ketones observed in WSD1 overexpressing lines has raised the question about how the increased expression of WSD1, a component of acyl-reduction pathway, affects the decarbonylation pathway, and where those wax constituents are being synthesized. It has to be a sort of pathway connections or regulation that we are not aware of and requires further investigations. In an attempt to provide an answer to that question, we performed an in silico protein–protein interaction analysis by searching the STRING database (https://string-db.org/, accessed date: March 2020) of any known protein that may physically or functionally interact with the WSD1 enzyme (Appendix A). The search resulted in no direct interaction, except a candidate gene that co-expressed with WSD1, namely Cytochrome P450 (TAIR ID: AT1G57750), which is involved in the formation of secondary alcohols and ketones in stem cuticular wax, and that requires further investigation.

## 4. Materials and Methods

For complete details of Materials and Methods, see Supporting Information.

### 4.1. Generation of Arabidopsis and Camelina Transgenics and Growth Conditions

The *Arabidopsis thaliana* (ecotype: Columbia) and *Camelina sativa* (cultivar: Suneson) transgenic lines were generated by introducing the Arabidopsis *WSD1* gene using the gene cassette *CaMV35S-p:WSD1:Actin2-t* and this cassette was cloned into the plant expression vector modified *pCAMBIA-DsRed* (Appendix A). This vector was used to transform Arabidopsis and Camelina using the *Agrobacterium*-mediated flower-dip method according to [36] using a protocol modified from [41,42]. T3 homozygous transgenic Arabidopsis and Camelina plants were further used in the subsequent analyses.

### 4.2. PCR Genotyping and qRT-PCR for Gene Expression Analysis

Total RNA was extracted from leaf tissue using the RNeasy mini kit (Sigma-Aldrich, St. Louis, MO, USA), and then cDNA was synthesized using Verso cDNA synthesis kit (Thermo Scientific, Waltham, MA, USA). The qRT-PCR was performed following the instructions for the Mastercycler ep realplex (Eppendorf, Hamburg, Germany), using gene-specific primers with SYBR green mix kit (Thermo Fisher Scientific) according to the manufacturer’s protocol.

### 4.3. Differential Regulation of WSD1 under Abiotic Stress Treatments

Arabidopsis WT seeds were surface-sterilized as described in [43]. The sterilized seeds were germinated and grown on a nylon screen placed on top of ½ strength solid MS media with vitamins (PhytoTechnology Laboratories, Lenexa, KS, USA), supplemented with 0.8% (*w/v*) Phytoblend agar (Caisson Laboratories, Smithfield, UT, USA), and 1% (*w/v*) sucrose, in Petri dishes for two weeks. Arabidopsis seedlings were exposed to 100 mM D-mannitol, 100 mM NaCl, and 2.5 μM ABA in liquid culture. Plant shoot and root samples were harvested separately at 0, 6, 12, and 24 h time intervals; washed with deionized water, flash frozen in liquid nitrogen, and stored at −80 °C until RNA extraction and qRT-PCR analyses.

### 4.4. Plant Growth Assays for Mannitol, ABA, Drought, and Salt Tolerance

Stress tolerance/sensitivity assays were performed as described previously [44]. The WT and WSD1 transgenic seeds were germinated in ½ strength MS agar plates (with 1% sucrose) in the presence or absence ABA (1 uM), D-mannitol (100 mM), and sodium chloride (NaCl, 75 mM), and the seedlings were maintained for 21 days with a 16/8 h light/dark cycle at 22/18 °C day/night temperature. At the end of ~21 days, plants were photographed, and shoot fresh weight and root length were recorded.

The drought stress assay in soil was performed after the plants attained sufficient biomass (4-wk-old seedling); the pots were saturated with tap water before starting the drought treatment by water withholding. Watering was withheld for 12 days until plants showed signs of wilting. The soil moisture was recorded daily during this period. Subsequently, the plants were re-watered to recover from stress and were analyzed for signs of permanent damage and photographed. The salt stress assay was performed on ~28-day-old plants grown in 4.0-inch pots. The pots were saturated with either tap water to serve as the control or with 100 mM NaCl solution in trays for 6 days. Following the 6-day treatment, the pots were placed in trays filled with tap water for several days for recovery.

For drought stress in Camelina in greenhouse conditions, seeds were germinated in potting mix soil and then 2-week-old seedlings were transferred to sand pots (5-inch width and 16-inch length) and filled with an equal weight of sand. After attaining sufficient biomass, the pots were saturated with tap water before the drought treatment was applied by water withholding. The 5-week-old plants were then exposed to 50% and 75% reductions in soil moisture and soil moisture was recorded daily during this period. The signs of wilting were recorded in both WT and WSD1 expressing plants after 2 and 4 weeks of treatment. Under control (well-watered) and drought conditions, the fresh biomass and the number of branches in Camelina plants were recorded, and at seed maturation, the seed mass and seed yield were determined as described in [36].

### 4.5. Cuticular Wax Loading and Composition Analysis

Cuticular waxes were extracted from Arabidopsis leaves (~500 mg) and stems (~200 mg) of 4-week-old plants in chloroform (~5 mL) for 30 s at room temperature, following the method described by [30]. The extracts were subjected to qualitative and quantitative analyses for cuticular waxes using gas chromatography with mass spectrometry (GC/MS) for identification, followed by gas chromatography with flame ionization detection (GC-FID) for quantification. Detailed information for wax loading and composition analysis is provided in the Appendix A.

### 4.6. Scanning Electron Microscopy (SEM)

Cryogenic SEM (ZEISS Inc., Oberkochen, Germany) was used to view epicuticular wax crystallization patterns. Inflorescence stem segments from tip to 3 cm and the fourth rosette leaves were collected from Arabidopsis wild-type (Col-0) and WSD1 overexpressing plants after four weeks of growth, as described previously [30,45].

### 4.7. Leaf Water Loss Assays and Chlorophyll Leaching Assays

The 4-week-old Arabidopsis plants grown in soil under growth chamber conditions were used for the leaf water loss assay following the methods described in [30]. The chlorophyll content in Arabidopsis leaves was determined as described in [30].

### 4.8. Statistical Analysis

The number of replicates (*n*) and the standard error (SE) are shown for most measurements. The data were analyzed with the SAS version 9.1 (www.sas.com) using ANOVA (*p* < 0.05) on the corresponding degrees of freedom (df), followed by Dunnett’s procedure for pairwise comparisons of all treatments in transgenic plants compared to the nontransgenic WT control.

## Figures and Tables

**Figure 1 ijms-22-05173-f001:**
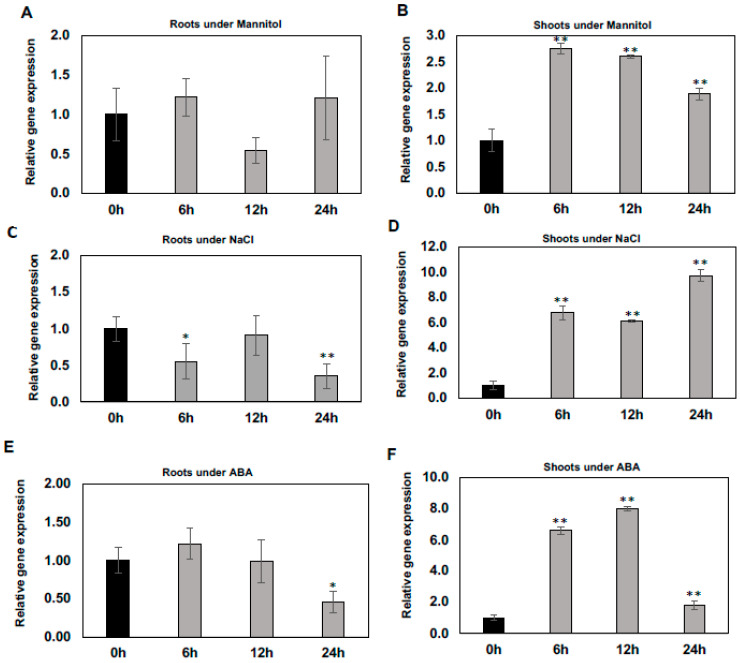
Regulation of *WSD1* gene under abiotic stresses. Two-week-old Arabidopsis plants were treated with stress conditions, including Mannitol and NaCl, and the signaling hormone ABA at the indicated time points (0, 6, 12, and 24 h). Results shown here are the effects of Mannitol on *WSD1* gene expression in roots (**A**) and shoots (**B**); effects of NaCl on *WSD1* gene expression in roots (**C**) and shoots (**D**); and effects of ABA on *WSD1* gene expression in roots (**E**) and shoots (**F**). Gene expression levels were normalized with respect to the internal control *elongation factor 1α* (*EIF1α*) gene. Error bars represent the mean ± SE levels of the relative abundance of three biological replicates. A statistical analysis was performed using Student’s *t*-test (** *p* < 0.01 and * *p* < 0.05).

**Figure 2 ijms-22-05173-f002:**
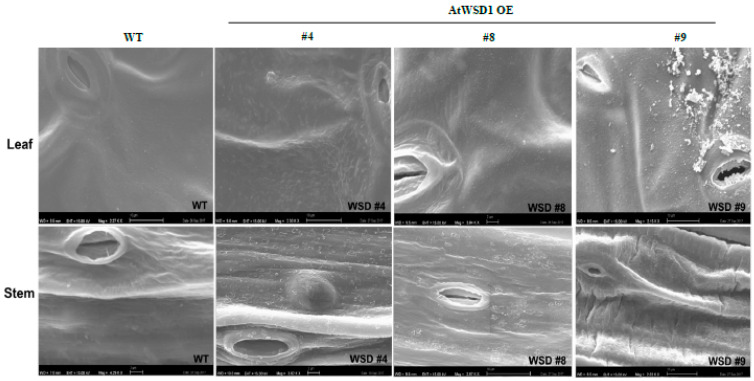
Scanning Electron Microscopy (SEM) showing the effects of WSD1 overexpression on epicuticular wax deposition in Arabidopsis leaf and stem tissues. Four-week-old Arabidopsis plants expressing the *CaMV 35s::WSD1* gene cassette and WT grown in soil were examined for the epicuticular wax crystals by scanning electron microscopy. Bars = 2 μm.

**Figure 3 ijms-22-05173-f003:**
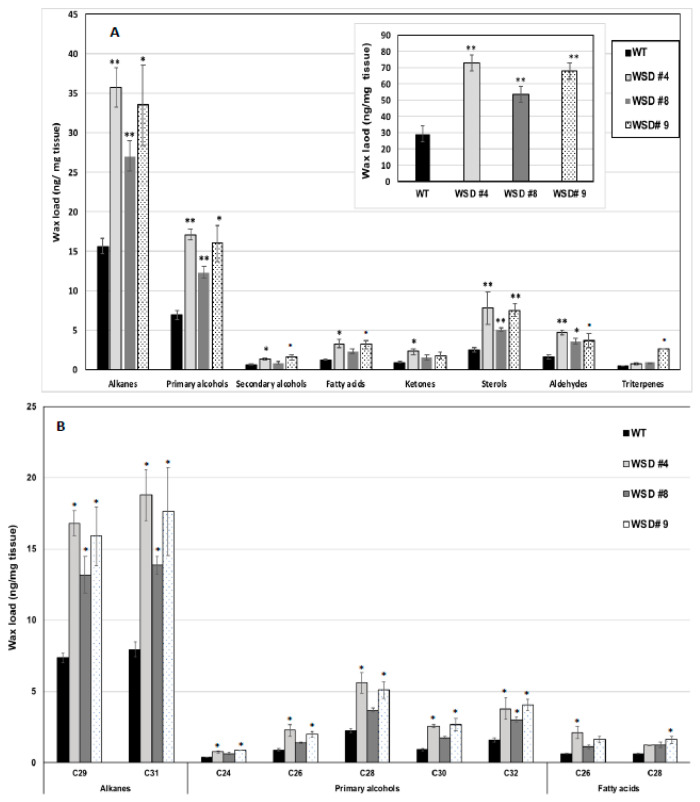
Cuticular wax loading and composition in WSD1 transgenic and WT plants obtained from Arabidopsis leaf tissues. Total wax loads and coverage (**A**) and carbon chain length (**B**) of individual compound classes are given with standard error (*n* = 3). Rosette leaves of 4-week-old wild type (Col-0, WT) and WSD1 overexpressing lines grown in soil were used for analysis of cuticular wax loads. Three biological replicates were averaged and statistically analyzed using a Student’s *t*-test (** *p* < 0.01 and * *p* < 0.05). Bars indicate the mean values (ng per mg FW) ± standard error (*n* = 3).

**Figure 4 ijms-22-05173-f004:**
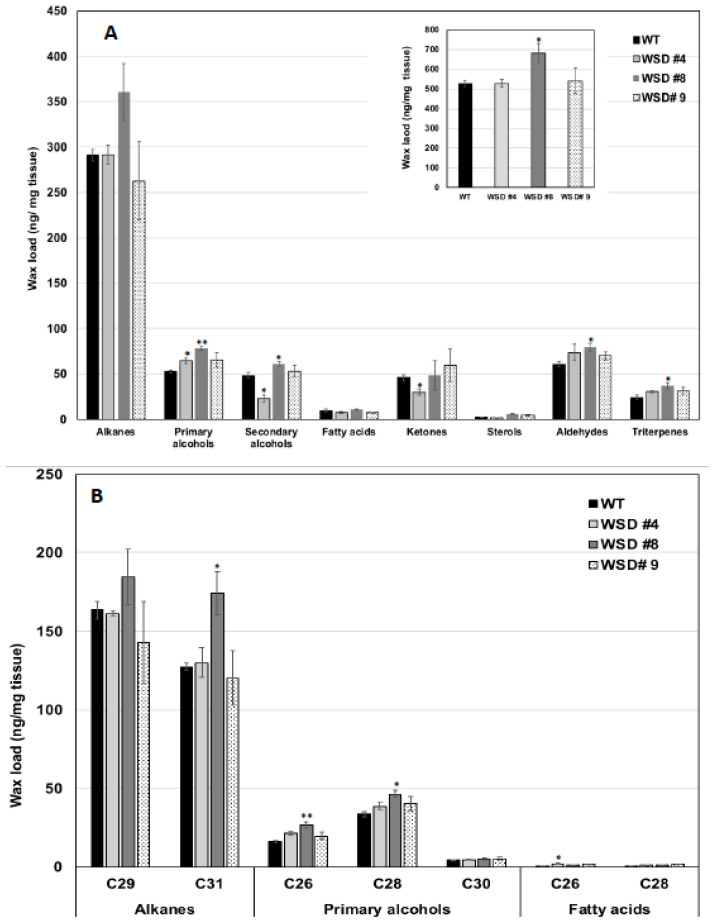
Cuticular wax loading and composition in WSD1 transgenic and WT plants obtained from Arabidopsis stem tissues. Total wax loads and coverage (**A**) and carbon chain length (**B**) of individual compound classes are given with standard error (*n* = 3). Rosette leaves of 4-week-old wild type (Col-0, WT) and WSD1 overexpressing grown in soil were used for analysis of cuticular wax loads. Three biological replicates were averaged and statistically analyzed using a Student’s *t*-test (** *p* < 0.01 and * *p* < 0.05). Bars indicate the mean values (ng per mg FW) ± standard error (*n* = 3).

**Figure 5 ijms-22-05173-f005:**
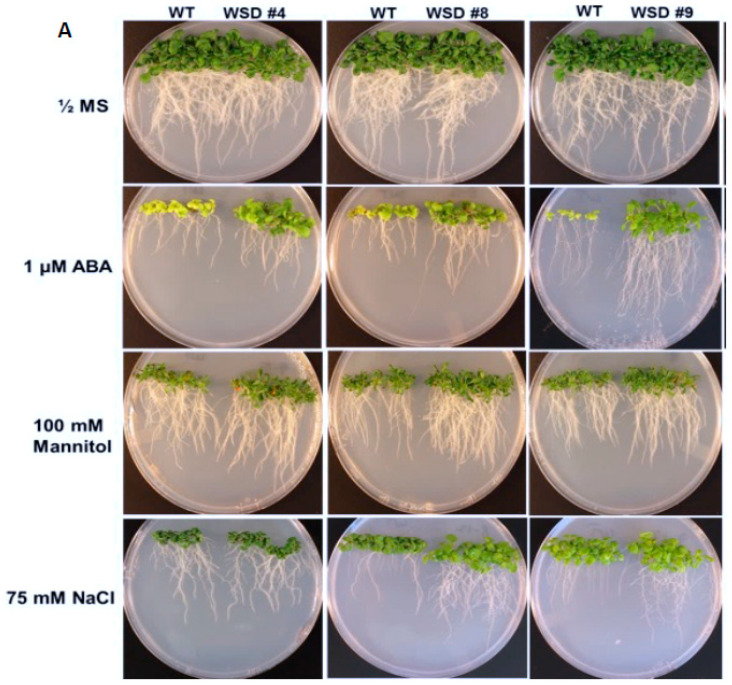
Screening WSD1 transgenics for tolerance to ABA, Mannitol, and salt stresses. WT and WSD1 transgenic plants were grown on ½ MS medium alone or supplemented with ABA (1 uM), Mannitol (100 mM), and NaCl (75 mM) for 21 days. (**A**). The transgenics are shown as WSD #4, 8, 9 and WT represent non-transgenic control plants. (**B**,**D**,**F**,**H**) represent root length (cm) measurements under control, ABA, Mannitol, and NaCl treatments, respectively. (**C**,**E**,**G**,**I**) represent plant total biomass (gm) under control, ABA, Mannitol, and NaCl treatments, respectively. Four biological replicates were averaged. The average and standard deviation values are represented for four replicates of 12 seedlings each for WT and all WSD1 transgenic lines. The asterisks represent significant difference in biomass accumulation and root length compared with WT plants, * *p* < 0.05, ** *p* < 0.01.

**Figure 6 ijms-22-05173-f006:**
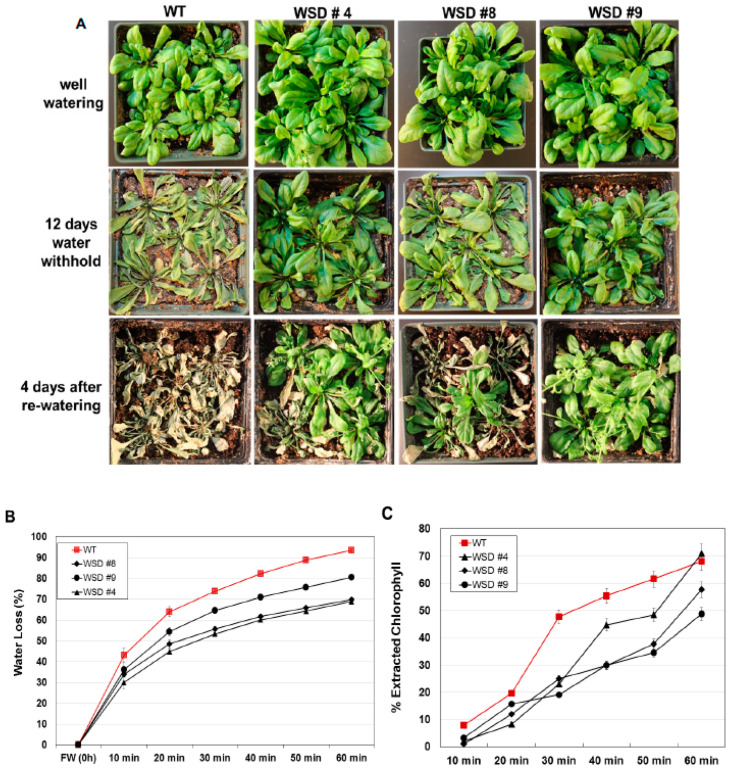
WSD1 overexpressing transgenic plants exhibit improved drought-stress tolerance in Arabidopsis. Phenotypic changes in WSD1 transgenics and WT (Col-0) plants subjected to water withholding. Water was withheld from 4-wk-old Arabidopsis plants for 12 days before watering was resumed. (**A**) Arabidopsis plants maintained under normal watering conditions (upper panel), 12 days after water withholding (middle panel), and 4 days after re-watering (lower panel). (**B**) Cumulative transpirational water loss and (**C**) Chlorophyll leaching assays from desiccated Arabidopsis leaves of the WT and WSD1 overexpressing lines (WSD #4, 8, and 9). Data in (**B**,**C**) represent the mean of 5 independent measurements, and the bars represent standard error (±SE, *n* = 5).

**Figure 7 ijms-22-05173-f007:**
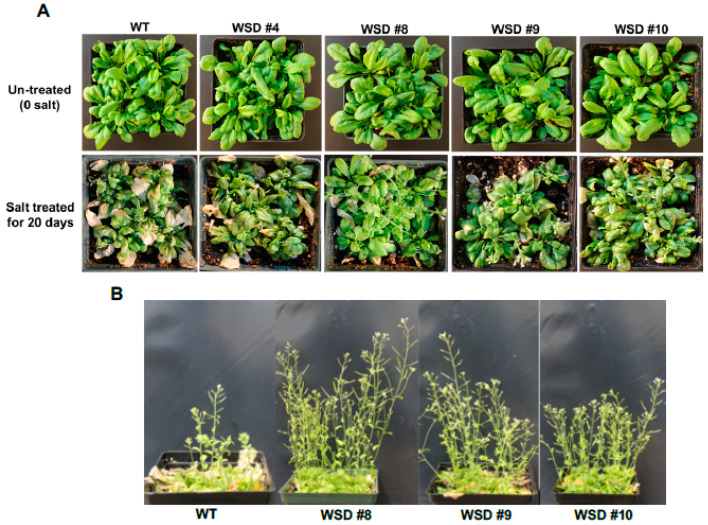
WSD1 overexpressing transgenic plants exhibit improved salinity-stress tolerance in Arabidopsis. Phenotypic changes in Arabidopsis transgenics and WT (Col-0) plants subjected to salt treatment. The images represent 4-wk-old Arabidopsis WT and transgenic plants either untreated (0 mM salt) or treated with water supplemented with 100 mM NaCl (**A**) and ~ 6-wk plants recovered from salt stress (**B**).

**Figure 8 ijms-22-05173-f008:**
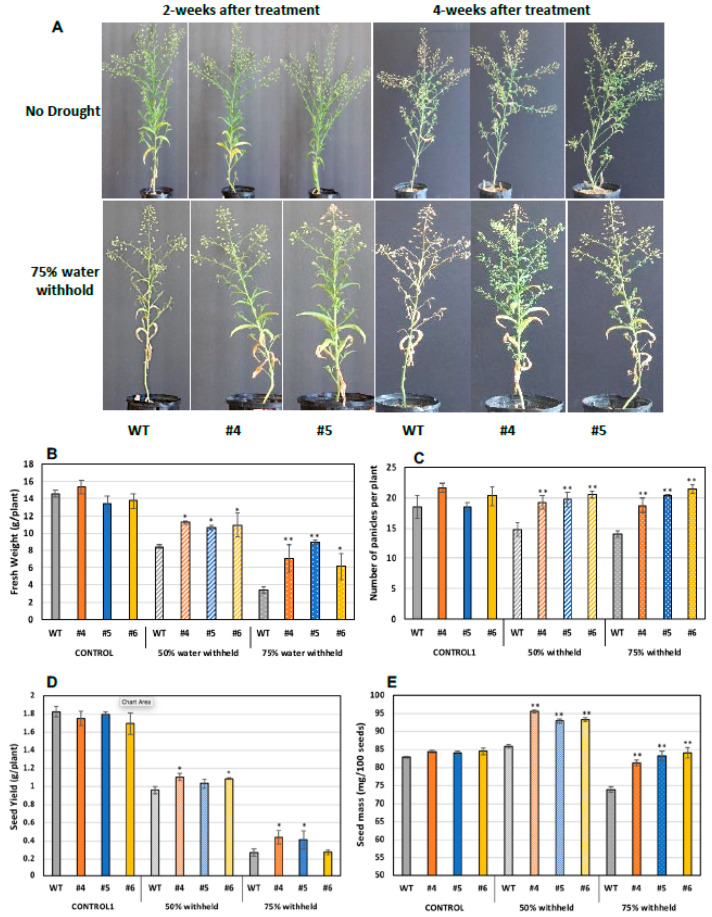
Transgenic Camelina lines overexpressing WSD1 exhibit improved drought-stress tolerance. Phenotypic changes in WSD1 transgenic Camelina lines (WSD #4 and 5) and WT plants subjected to water withholding. Moreover, 50% and 75% water was withheld from 4-wk-old Camelina plants till maturity. (**A**) phenotype of Camelina plants maintained under normal watering conditions (upper panels), 2 and 4 weeks after water withholding (lower panels). (**B**) Total fresh biomass (g/plant), (**C**) total number of panicles per plant, (**D**) average seed yield (g/plant) and (**E**) The average seed weight (mg/100 seeds) of Camelina lines overexpressing WSD1 (WSD #4, 5, and 6) and WT controls. Data represents the mean of 4 independent plants of each line, and the bars represent Standard error (±SE, *n* = 4). Asterisks denote significance of differences between WT and transgenics (Student’s t-test): ** *p* < 0.01; * *p* < 0.05.

## Data Availability

All data supporting our findings are contained in the additional file, which have been provided as Appendix A.

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
