# Peer review of "Increased Cuticle Waxes by Overexpression of WSD1 Improves Osmotic Stress Tolerance in Arabidopsis thaliana and Camelina sativa"

_ijms, 2021, doi:10.3390/ijms22105173_

Round 1

Reviewer 1 Report

I have two main concerns: (1) the accumulation level of WSD1 protein should be estimated (Western blot) and (2) gas exchange parameters should be measured (CO2 assimilation and stomatal conductance as well as stomatal transpiration). 

Reviewer 2 Report

This manuscript is describing the characterization of Arabidopsis and Camelina plants overexpressing a bifunctional wax synthase/acyl-CoA:diacylglycerol acyltransferase (WSD1) gene, which plays a critical role in wax ester synthesis. These transgenic plants confer abiotic stress resistance including ABA, mannitol, drought, and salinity. Transgenic plants are also able to recover from drought and salinity better than the WT plants, due to their reduced cuticular transpirational rates and cuticle permeability as well as reduced chlorophyll leaching than the WT. This manuscript might be accepted for publication in International Journal of Molecular Sciences after the following points would be addressed.

1. In Figure 5H and 5I, the authors showed that WSD1 transgenic plants, when treated with salt (75 mM NaCl), the root length and shoot biomass of only line #8 were significantly increased as compared to that in WT plants. Although the qRT-PCR results confirmed that the WSD1 gene was constitutively overexpressed at higher levels in transgenics, line #8 was not the highest levels among four lines of transgenics (Supplemental Figure S1B, C ). The authors presented some interpretations of the impact of WSD1 overexpression in conferring tolerance to abiotic stress in Discussion. However, it would be desirable to discuss the reason why only line #8 was significantly increased salinity stress tolerance.

2. I think Discussion is relatively redundant and might be reduced.

Author Response

Reviewer # 2

This manuscript is describing the characterization of Arabidopsis and Camelina plants overexpressing a bifunctional wax synthase/acyl-CoA:diacylglycerol acyltransferase (WSD1) gene, which plays a critical role in wax ester synthesis. These transgenic plants confer abiotic stress resistance including ABA, mannitol, drought, and salinity. Transgenic plants are also able to recover from drought and salinity better than the WT plants, due to their reduced cuticular transpirational rates and cuticle permeability as well as reduced chlorophyll leaching than the WT. This manuscript might be accepted for publication in the International Journal of Molecular Sciences after the following points would be addressed.

1. In Figure 5H and 5I, the authors showed that WSD1 transgenic plants, when treated with salt (75 mM NaCl), the root length and shoot biomass of only line #8 were significantly increased as compared to that in WT plants. Although the qRT-PCR results confirmed that the WSD1 gene was constitutively overexpressed at higher levels in transgenics, line #8 was not the highest levels among four lines of transgenics (Supplemental Figure S1B, C ). The authors presented some interpretations of the impact of WSD1 overexpression in conferring tolerance to abiotic stress in Discussion. However, it would be desirable to discuss the reason why only line #8 was significantly increased salinity stress tolerance.

Response:

As seen in Figure 5, as per the reviewer’s comment, there are variations between the four lines tested when the seedlings of both WT and WSD1 lines were exposed to NaCl treatment. The biomass data shown in the graph are on the average basis of at least three independent plate/assay. The results showed only line #8 exhibits salinity tolerance phenotype, but this is not completely true as all the three lines tested (WSD# 8, 9, 10) were tolerant to higher levels of salinity stress (100 mM NaCl) in soils as shown in Figure 7.    

The fact that the salinity tolerant phenotype was obvious only in L#8 and not in the other lines could be due to the condition of the assay itself since the seedlings on MS medium were exposed to 75 mM NaCl for ~ 21 days, and no change in medium or NaCl concentrations, which could be considered “mild” stress conditions. However, in soils where we exposed the seedlings to salinity stress, the addition of NaCl was repeated every time the plants were watered which elevated the concentrations of NaCl in the soils, thus causing much severe stress conditions and all transgenic lines showed salinity tolerance phenotypes. Some discussion on salinity tolerance phenotypes observed in our WSD1 transgenics is given in lines 333-361 in the revised manuscript.

Since the main reason that we selected drought and salinity stresses tolerance aspects of WSD1 herein is to provide evidence that WSD1 may have multiple stress-related functions, and more investigations for the salinity stress attributes will be the topic of our future manuscripts in Arabidopsis and Camelina.   

2. I think Discussion is relatively redundant and might be reduced.

Response: As per the reviewer’s suggestion, we have revised the discussion section and reduced the redundancy which can be seen in the track-0change copy.

Round 2

Reviewer 1 Report

No my suggestions were included into the manuscript. Thus, I cannot accept the revision.